# NERvous About My Health: Constructing a Bengali Medical Named Entity Recognition Dataset

**Alvi Aveen Khan[1], Fida Kamal[1], Nuzhat Nower[1],**
**Tasnim Ahmed[1,2], Sabbir Ahmed[1],** and **Tareque Mohmud Chowdhury[1]**
[1]Islamic University of Technology      [2]Queen's University
[1]{alviaveen, fidakamal, nuzhatnower, sabbirahmed, tareque}@iut-dhaka.edu
[2]tasnim.ahmed@queensu.ca

## Abstract

The ability to identify important entities in a text, known as Named Entity Recognition (NER), is useful in a large variety of downstream tasks in the biomedical domain. This is a considerably difficult task when working with Consumer Health Questions (CHQs), which consist of informal language used in day-to-day life by patients. These difficulties are amplified in the case of Bengali, which allows for a huge amount of flexibility in sentence structures and has significant variances in regional dialects. Unfortunately, the complexity of the language is not accurately reflected in the limited amount of available data, which makes it difficult to build a reliable decision-making system. To address the scarcity of data, this paper presents 'Bangla-HealthNER', a comprehensive dataset designed to identify named entities in health-related texts in the Bengali language. It consists of 31,783 samples sourced from a popular online public health platform, which allows it to capture the diverse range of linguistic styles and dialects used by native speakers from various regions in their day-to-day lives. The insight into this diversity in language will prove useful to any medical decision-making systems that are developed for use in real-world applications. To highlight the difficulty of the dataset, it has been benchmarked on state-of-the-art token classification models, where BanglishBERT achieved the highest performance with an F1-score of $56.13 \pm 0.75\%$. The dataset and all relevant code used in this work have been made publicly available[1].

## 1 Introduction

A fundamental component of nearly all automated Question Answering (QA) systems is a Named Entity Recognition (NER) model (Nadapana and Kommanti, 2022; Aliod et al., 2006). To accu-rately answer a question, the system must be capable of identifying important entities in the text. This is even more critical when addressing health-related queries from the general public, known as Consumer Health Questions (CHQs). CHQs are often difficult to answer due to the informal language used (Mishra and Jain, 2016), and most patients only provide a vague description of their issues. This presents a unique challenge to current systems, most of which were developed for use with medical documents that use formal language and concrete scientific terms to describe health issues (Carini et al., 2021). Being able to extract the correct medical knowledge required to answer the question from the vague descriptions containing unstandardized medical terms used by the general public is a challenging task.

The difficulties described are further exacerbated in the Bengali language, which poses greater challenges compared to English for several reasons. Bengali allows for a significantly larger number of inflections, with 220 inflections existing in total compared to just 9 for English (Bhattacharya et al., 2005). This forces the models to learn a large number of words that are nearly identical but have slightly different meanings. For example, in the sentence 'I have fallen ill', the verb 'fallen' remains unchanged even if the subject is changed to 'You' or 'She'. In Bengali, however, the same sentence ('আমি অসুস্থ হয়ে পড়েছি।') will change the verb depending on the subject: 'পড়েছি' for the first person, 'পড়েছো' for the second-person, and 'পড়েছে' for the third-person. There is also greater flexibility regarding sentence structuring. Whereas the order of words is relatively rigid in English, and breaking the order makes sentences grammatically incorrect, the same is not true for Bengali. For example, 'I have a headache' is correct in English, but 'headache have I' sounds odd and is unlikely to be used. However, the Bengali versions of these sentences ('আমার মাথা ব্যথা' and 'মাথা ব্যথা আমার' re-

---

[1]https://github.com/alvi-khan/Bangla-HealthNER

spectively) are both correct and commonly used. Furthermore, there is a large amount of diversity in the language as there are numerous dialects that are so significantly different from each other that people from one region are often entirely unable to comprehend the language of another (Shahed, 1993). These attributes make Bengali an incredibly complex language with a huge amount of variation and intricacies.

Unfortunately, the amount of data available for Bengali, especially in a medical context, is severely limited and fails to reflect the complex nature of the language. This makes it challenging to create a system that accurately understands the language in the context of CHQs. To address this issue, we present 'Bangla-HealthNER', the largest human-annotated Bengali medical NER dataset comprising 31,783 samples. The data has been collected from a public online health platform in Bangladesh, ensuring that it accurately captures the variety and complexity of the language used by native Bengali speakers in their day-to-day lives. The samples also consist of a large amount of text that combines Bengali and English, using words from both languages or transliterating words from one language using alphabets from the other, a common occurrence in informal conversations known as 'code-switching'. The large variety in language present in the dataset should allow models to gain a thorough understanding of medical terminology in informal contexts.

## 2 Literature Review

Although there is a significant amount of data available for general Bengali NER (Haque et al., 2023; Chowdhury et al., 2018; Shahgir et al., 2023), the availability of data specifically tailored for Bengali medical NER remains limited. Currently, there are two datasets publicly available: one introduced by Islam et al. (2022), which we will be referring to in this paper as the 'Telemedicine Dataset', and the 'BanglaBioMed' dataset (Sazzed, 2022). Both of these datasets have relatively small sizes, with the Telemedicine dataset consisting of 190 samples and nine entity labels and BanglaBioMed consisting of 1100 sentences and four entity labels. This makes the datasets unsuitable for modern transformer-based architectures, which rely on large amounts of data for effective learning. Additionally, the data for both datasets were collected from sources that use

| Metric | Count |
|---|---|
| Sample count | 31,783 |
| Sentences | 144,136 |
| Percentage of entity tokens | 23.47% |
| Avg. words per sample | 175.81 |

Table 1: Statistical metrics of Bangla-HealthNER

formal language, differing greatly from the language people use in their day-to-day speech. The telemedicine dataset was compiled by manually transcribing telephone conversations with patients, while BanglaBioMed was created by extracting text from health articles published in a popular national newspaper.

The small dataset sizes combined with their formal nature means that systems trained on the existing datasets will be unable to perform well in real-world use cases. The introduction of Bangla-HealthNER addresses both of these limitations since it is larger in size and also accommodates much more variety in language.

## 3 The Bangla-HealthNER Dataset

### 3.1 Data Collection

The dataset was collected from a popular Bengali online medical health platform[2], which provides publicly available health-related queries from users as well as answers by medical professionals. Both the questions and answers were collected and are considered as separate samples in the dataset. Samples were collected via random sampling in order to obtain an unbiased representation of the data. An overview of various metrics for the dataset and the different entities is presented in Table 1 and Fig. 1, respectively. Compared to existing datasets, Bangla-HealthNER not only has significantly more samples but also much longer samples, which mostly consist of multiple sentences. Lengthier training data will force systems to learn to retain contextual information for longer distances to be able to accurately identify entities.

### 3.2 Pre-Processing

The dataset was first cleaned by removing duplicate entries, URLs, and spam text. Afterwards, personally identifiable information was removed.

---

[2]https://daktarbhai.com/

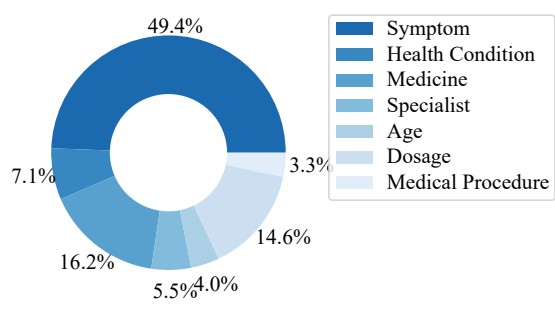

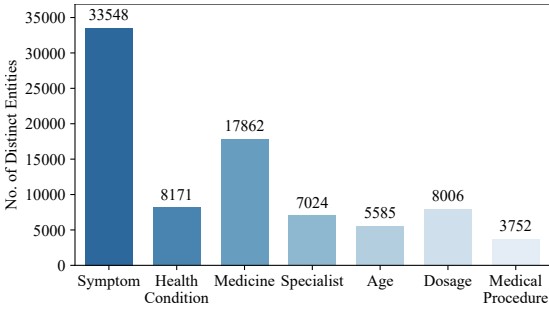

(a) Distribution of entities types

(b) Comparison of distinct number of entities per type

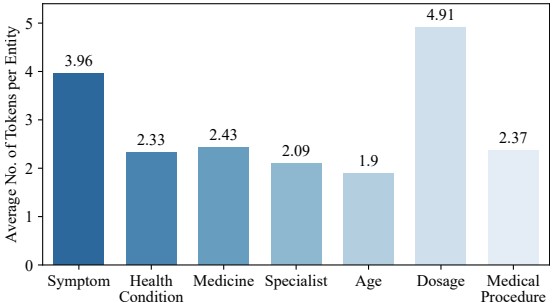

(c) Comparison of the average number of tokens per entity for each entity type

Figure 1: Statistics for individual entities of Bangla-HealthNER

| Entity | Description |
|---|---|
| Symptom | Indicators of an underlying issue, such as pain (ব্যথা), fever (জ্বর), etc. |
| Health Condition | Diseases like sinusitis (সাইনুসাইটিস) or health events like pregnancy (প্রেগনেন্সি). |
| Age | Numbers (numeric or textual) that specifically convey the age of a patient, e.g., 5 months (পাঁচ মাস), 2 years (২ বছর), etc. |
| Medicine | Names of medicine like Omidon, Paracetamol (প্যারাসিটামল), etc. |
| Dosage | The amount of medication to take, such as 1 tablet a day, 2 spoonfuls of syrup, etc. |
| Medical Procedure | Medical tests like Ultrasound (আল্ট্রাসাউন্ড) or procedures like surgery (সার্জারি). |
| Specialist | Names of specific medical specialists like Medicine Specialist (মেডিসিন বিশেষজ্ঞ), Urologist (ইউরোলোজিষ্ট), etc. |

Table 2: Entity descriptions for Bangla-HealthNER

Names were removed with the help of a general Bengali NER model[3], followed by manual inspection, and email addresses and phone numbers were removed using regular expressions.

### 3.3 Annotation

The dataset was annotated using an open-source annotation tool[4] by a team of 20 undergraduate students with at least a higher secondary level of formal education in biology. They were provided with extensive guidelines to follow during the an-

notation process, the details of which are provided in Appendix A.2. The annotations follow the Inside-Outside-Bagging (IOB) format with non-overlapping entities and consist of seven types of entities as described in Table 2. Five percent of the samples given to the annotators were taken from a pre-annotated subset of the dataset and were used to calculate the Inter-Annotator Agreement (IAA). The guidelines, entity definitions, and pre-annotated data were all carefully prepared by the authors after extensively studying several works related to the domain (Li et al., 2016; Sazzed, 2022; Bodenreider, 2004). The quality of the fi-

[3]https://pypi.org/project/bnlp-toolkit
[4]https://github.com/tecoholic/ner-annotator/

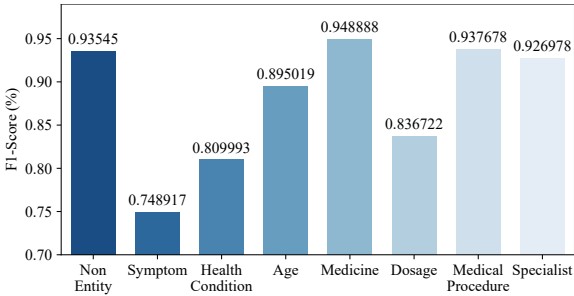

Figure 2: Average IAA F1-Score (non-weighted) per entity type

nal dataset was also ensured by following the work of Kabir et al. (2023). The average weighted F1-score for IAA was found to be 88.56%, and the average Cohen's Kappa score (Cohen, 1960) was found to be 67.19% (substantial agreement). An entity-wise breakdown of the average IAA F1-Score is provided in Fig. 2.

The annotation process revealed certain issues with the existing tagging formats used for NER tasks. For example, consider the phrase 'হাতে তীব্র ব্যথা', which can be translated word-for-word into 'arm severe pain'. According to our guidelines, the intensifier word 'তীব্র' should not be considered part of the symptom so that the accurate tags would be [B-Symptom, O, I-Symptom]. Unfortunately, the IOB format does not allow a single entity to be split into two parts in this manner. This issue does not occur in English datasets since adjectives nearly always come before or after nouns in the English language, which is why the word-for-word translation is not grammatically correct. The grammatically correct version of this phrase, 'severe arm pain', would have the tags [O, B-Symptom, I-Symptom]. As a consequence of this shortcoming of the tagging format, the annotators were frequently forced to deviate slightly from the guidelines and include the non-entity tokens that appeared in between entity tokens. The issue was particularly pronounced for the 'Symptom' and 'Health Condition' entities since the informal nature of the data meant that patients frequently described issues in a verbose manner.

It was also found that there were subtle variations in very similar concepts, which should create challenging situations. For example, the phrase 'মাথা ব্যথার ওষুধ' ('headache medicine') should be tagged as [B-Medicine, I-Medicine, I-Medicine] whereas the phrase 'মন্টেয়ার ওষুধ' ('Montair medicine') should be tagged as [B-Medicine,

O]. This distinction occurs because the name of the medicine, 'Montair', in the second case implies that it is a medicine, thus making the word 'medicine' redundant, unlike in the first case. Similarly, the phrase 'নাক কান গলা ডাক্তার' ('ENT Doctor') which should be tagged as [B-Specialist, I-Specialist, I-Specialist, I-Specialist] but the phrase 'নাক কান গলা বিশেষজ্ঞ ডাক্তার' ('ENT Specialist Doctor') should be tagged as [B-Specialist, I-Specialist, I-Specialist, I-Specialist, O]. Again, the word 'doctor' in the second case is redundant since it is implied by the words 'ENT Specialist'.

## 4 Benchmarking

To establish a benchmark, the dataset was fine-tuned on three token classification models-BanglaBERT (Bhattacharjee et al., 2022), BanglishBERT (Bhattacharjee et al., 2022), and mBERT (Devlin et al., 2019). Aside from this, to analyze the performance of large language models on Bengali medical NER, we also included the inference results of zero-shot ChatGPT (gpt-3.5-turbo, accessed June 2023) (Brown et al., 2020).

For the benchmarking process, the dataset was divided into training, validation, and test splits using the ratio $80 : 10 : 10$. Further details about the experimental setup are provided in Appendix A.1, and Table 3 provides the results of our experiments. The scores for the evaluation metrics are calculated at the token level and are averaged across multiple runs except in the case of GPT 3.5.

Both BanglaBERT and BanglishBERT outperformed the mBERT model, which reinforces the understanding that models pre-trained on a specific language achieve better performance on downstream tasks than multilingual ones. BanglishBERT achieved the highest scores, likely due to the presence of text that combines Bangla and English in the dataset.

The GPT 3.5 model was not fine-tuned on our dataset but was given prompts to assign labels to each word in the test samples. Past literature studying the performance of ChatGPT on downstream tasks related to bio-medicine without fine-tuning found that domain-specific models achieve significantly better results, even if expert prompting is used (Jahan et al., 2023). Our results agree with this conclusion, as all the fine-tuned models outperform zero-shot ChatGPT by a large margin.

A thorough error analysis of the predictions of the fine-tuned models reveals that the expected

| Model | Accuracy (%) | Precision (%) | Recall (%) | F1-Score (%) |
|---|---|---|---|---|
| GPT 3.5 (zero-shot) | 72.43 | 28.62 | 10.64 | 15.51 |
| mBERT | $87.98 \pm 0.25$ | $49.46 \pm 1.15$ | $61.90 \pm 0.47$ | $54.98 \pm 0.82$ |
| BanglaBERT | $87.90 \pm 0.01$ | $48.00 \pm 0.38$ | $63.66 \pm 0.29$ | $54.74 \pm 0.34$ |
| BanglishBERT | $88.42 \pm 0.14$ | $50.20 \pm 0.62$ | $63.65 \pm 0.95$ | $56.13 \pm 0.75$ |

Table 3: Benchmark results for Bangla-HealthNER

challenging situations discussed in section 3.3 did indeed cause issues. The models seemed to have difficulty understanding subtle differences in similar concepts. For example, a large number of the responses to questions concluded with advice to visit a specialist, such as 'একজন এ.এন.টি. বিশেষজ্ঞ দেখান।' ('Please visit an E.N.T. specialist.'). It was observed that samples that concluded slightly differently, such as 'বারডেম হাসপাতালের শিশু বিভাগে যোগাযোগ করুন।' ('Please contact the children's unit of BIRDEM Hospital') resulted in the models marking incorrect phrases, 'children's unit' in this case, with the label 'Specialist'. Similar issues were also seen with the 'Age' class, where the models marked any mention of time in years, such as '৩ বছর আগে হাত ভেঙে গিয়েছিলো।' ('My arm broke 3 years ago') with the label 'Age'. This was most likely caused by the majority of such mentions in the dataset genuinely being the patient's age.

## 5   Conclusion

In this paper, we introduced a manually annotated Bengali Named Entity Recognition dataset of health-related texts. The paper highlights several key features of the dataset, including its ability to capture the linguistic diversity and complexity of the Bengali language, as well as its emphasis on the informal nature of speech used by the average patient in medical contexts. We evaluated the dataset on Bengali and multilingual token classification models to establish a benchmark and further explored the performance of the large language model GPT 3.5 on the dataset. Given that the dataset introduces significant variety in language to the domain compared to existing data, architectures trained on past data struggled to perform well. This opens up a new direction for future work, which can concentrate on exploring methods to improve this performance.

The research and data presented in this paper have implications that extend beyond the domain of NER in CHQs. The challenges addressed, such as dealing with informal language, complex sentence structures, and regional dialects, are observed in various downstream tasks, meaning our findings can be generalized to broader contexts and applications. The dataset can also be leveraged to achieve cross-domain adaptability in fields such as social media, customer reviews, and online forums, where users communicate using colloquial and unstructured language. The proposed insights also have cross-linguistic transferability to other Indo-Aryan languages with similar linguistic characteristics. This could prove beneficial in developing NER models for languages with limited labelled data. The adaptability of NER as a feature can facilitate automated clinical decision-making systems as well.

## Limitations

One limitation that caused notable difficulties during the annotation process was the shortcomings of the tagging format used, as discussed in section 3.3. Despite our extensive guidelines, the unstructured nature of the Bengali language resulted in our annotators being forced to mark words that separated a single multi-word entity as part of the entity due to the limitations of the tagging format. This inevitably had a negative effect on the fine-tuning process. Working towards establishing a tagging format that addresses these limitations and fixing the annotations of this dataset accordingly is likely to lead to improved performance.

## Ethics Statement

The privacy of the patients of the health platform was a top priority during the data collection process. As such, the data was de-identified through manual inspection to ensure the removal of all personal information. All of the annotation work was undertaken either by the authors themselves or by hired individuals. The annotators were provided with monetary compensation for their work, which is above the minimum wage. The annotation process has also been de-identified to prevent any privacy violations of the annotators.

## Acknowledgements

The funding for this research was provided by the Islamic University of Technology (IUT) under the IUT Research Seed Grants (IUT RSG). We are grateful for their support.

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

## A Appendix

### A.1 Experiment Setup

The experiments done to establish our benchmark results were run on an Nvidia 3090 GPU with 24 GB of VRAM using the Trainer library available through Hugging Face and Cuda Version 11.6. The dataset was split into training, validation, and test sets at the ratio $80:10:10$ and trained for 25 epochs on each of the models. A batch size of 128 was used along with an AdamW optimizer, a linear learning rate scheduler, cross-entropy loss, 40% dropout, and a weight decay of 3e-2. A learning rate of 1e-4 was used with the first 20% of the training steps being used for warm-up. The use of Gradient Checkpointing additionally allowed us to use a maximum token length of 512.

In the course of our experiments, we also attempted to calculate an estimate of our carbon footprint. We did this by following the work of Lannelongue et al. (2021). It was found that each run of the experiments under our hardware setup generated approximately 331.17 grams, 222.68 grams, and 216.97 grams of CO2 for the mBERT, BanglaBERT, and BanglishBERT models respectively. It is interesting to see that the models that achieved comparatively better results in our experiments, BanglaBERT and BanglishBERT, also have a smaller impact on CO2 emissions.

### A.2 Annotation Guidelines

In addition to the basic instruction of labelling each word in the samples with one of the seven entity labels, the annotators were provided with extensive instructions on how to address a large variety of different scenarios that were seen to arise. These instructions are provided below. The colour code used for the examples is as follows: Symptom, Health Condition, Age, Medicine, Dosage, Medical Procedure, Specialist.

1. Entities should be annotated as specifically as possible. If an entity contains a sub-entity, the top-level entity should be annotated.
   Example:
   গত ৪ দিন ধরে মাথাব্যথা X
   গত ৪ দিন ধরে মাথাব্যথা ✓

2. A single entity should not be divided into two parts, even if this requires including some unnecessary information.
   Example:
   আমার মাথা গত ২ দিন ধরে অল্প ব্যথা করছে। X
   আমার মাথা গত ২ দিন ধরে অল্প ব্যথা করছে। ✓

3. Symptoms are indicators of a health condition. A health condition is a specific disease or medical event.
   Example:
   প্রায় সময় মাথাব্যথা হয়
   সাইনুসাইটিস হতে পারে
   হার্ট এটাক হয়েছে

4. Injuries are considered health conditions.
   Example:
   আমার পা মচকে গেছে

5. Unnecessary information, such as the severity of a symptom, should not be annotated.
   Example:
   মুখে ব্রণ উঠে অনেক বেশি X
   মুখে ব্রণ উঠে অনেক বেশি ✓

6. Redundant information (such as the word 'ডাক্তার' following the word 'বিশেষজ্ঞ') should not be annotated.
   Example:
   মাথা বেথার ওষুধ।
   মন্টেয়ার ওষুধ।
   নাক, কান, গলা ডাক্তার।
   নাক, কান, গলা বিশেষজ্ঞ ডাক্তার।

7. It is important to pay attention to contextual information when identifying entities. For example, the term 'Corona' can be used to describe both the COVID-19 disease and the COVID-19 pandemic in Bengali.
   Example:
   করোনা চলাকালীন ডাক্তার দেখতে পারছি না। X
   এগুলা কি কোরোনার লক্ষণ? ✓

8. Multiple symptoms mentioned together should be annotated separately.
   Example:
   bad cold and fever X
   bad cold and fever ✓

9. Medicine names can include the amount of chemical in each intake.
   Example:
   মোনাস ১০ মিগ্রা খান X
   মোনাস ১০ মিগ্রা খান ✓

10. Doctors frequently write the dosage in the format '0+1+1'.
    Example:
    মোনাস ১০ মিগ্রা 0+0+1

11. Medical specialists typically have uniquely identifiable names (e.g., গাইনেকোলজিস্ট) but are sometimes referred to by more generic names (e.g., গাইনির ডাক্তার).
    Example:
    একজন মেডিসিনের ডাক্তার দেখান