# OpenReview forum: "NERvous About My Health: Constructing a Bengali Medical Named Entity Recognition Dataset"
_EMNLP/2023/Conference — EMNLP 2023 Findings_

### Official Review · Reviewer_NwnB · 2023-07-31

**Soundness:** 3

**Excitement:**

4: Strong: This paper deepens the understanding of some phenomenon or lowers the barriers to an existing research direction.

**Paper Topic And Main Contributions:**

The article describes the creation and annotation of a Bengali dataset consisting of biomedical texts from an online consumer health platform. The authors describe the annotation procedure and conduct initial experiments with transformer-based BERT models and ChatGPT.

**Questions For The Authors:**

Q1: Can you give more details how the texts were selected from platform, e.g. random sampling or by using certain queries? (lines 150-155)

Q2: Why did you opt to divide the answers and the questions into separate samples (lines 154,155)? Can the connection still be reconstructed from the annotated dataset?  For better usability, it would be beneficial if the connection could be made.

Q3: Can you confirm that you have taken all legally necessary measures to anonymize the texts? In some countries, in addition to names, telephone numbers and email addresses, other secondary information (such as locations, age, etc.), which may be used to indirectly identify individuals, must be anonymized, too.

Q4: How are the results metrics of the benchmark computed – on token or on mention level? How is accuracy defined here? I guess accuracy on token level, right?


**Reasons To Accept:**

-	The work provides a resource for non-English language texts in the biomedical field. Resources in this field are very scarce and the dataset thus represents a valuable contribution for the research community.
-	The authors give a convincing motivation for the work and the article is essentially written in a comprehensible way (even if details are missing - see reasons to reject).


**Reasons To Reject:**

-	The paper misses to describe essential properties of the created dataset (e.g. number of (distinct) entities per type etc. – see detailed comment below), which complicates the assessment of the resource created
-	Details of the benchmark and the evaluation protocol are missing, which limits the reproducibility.
-	I am not entirely sure whether all necessary measures have been taken to ensure the anonymity of persons / authors of the texts collected.


**Reproducibility:**

4: Could mostly reproduce the results, but there may be some variation because of sample variance or minor variations in their interpretation of the protocol or method.

**Reviewer Confidence:**

4: Quite sure. I tried to check the important points carefully. It's unlikely, though conceivable, that I missed something that should affect my ratings.

**Typos Grammar Style And Presentation Improvements:**

-	Table 1 should be extended to include more detailed information about the created dataset, i.e number of (distinct) entities per entity type, average number of entities per text type (queston, answer), metrics concerning the lengths of the entities (e.g. average/median). This information help to get a better insight into the created dataset.
-  A small note: The shortcomings of the tagging format described in 3.3 could be circumvented by annotating the offsets in the text instead of IOB labels (and allowing an entity to span several offsets). However, this option is very rarely chosen and the number of approaches that support prediction of such non-consecutive entities seems even smaller to me.

---

> ### Author Rebuttal · Authors · 2023-08-28
>
> We extend our gratitude for the invaluable feedback you provided regarding our manuscript. We hope to address your concerns comprehensively in the subsequent text.
>
> **Code Reproduction**: We have provided details regarding the experimental setup and hyperparameters in Appendix A1. Moreover, all the codes related to dataset curation, preprocessing, post processing, baseline training, evaluation criteria, etc., will be made publicly available upon the acceptance of the manuscript. We will put extra effort into improving the codebase documentation as much as possible to ensure reproduction.
>
> **Anonymity measure**: As described in lines 168-173, the anonymization process involved the automated removal of personally identifiable information and was followed by a manual inspection by the annotators during the annotation process. The manual nature of the annotation process means that every sample was viewed by at least one human annotator at some point in time. We are confident that any personally identifiable information has definitely been caught in this process.
>
> **Text selection process from the platform**: The source of the data was from a publicly available Digital Health platform, and the texts were collected via random sampling. Specific queries were not used since our work concentrates on general healthcare and not specific subcategories.
>
> **Regarding treating the question and answer samples separately**: We opted to separate the questions and answers since our goal was to extract the NERs from a given statement regardless of it being a question/answer sample. Hence the connection between the Q-A was not preserved.
>
> However, since the reviewer suggests that retaining the connection would be beneficial, we can also release a version of the annotated dataset that retains the connection. Unfortunately, our benchmark results would not be applicable to that version.
>
> **Legal issues regarding Anonymity measure**: We have taken care to remove any and all information that is considered personally identifiable under the data privacy laws of Bangladesh.
>
> **Result metrics**: We have computed metrics for accuracy, precision, recall, and F1-score at the token level. The score reported in the abstract is for the F1-Score. We will revise the manuscript and remove any ambiguity regarding this issue.
>
> **About extending Table 1**: We appreciate the suggestion and will extend Table 1 to include this information. A portion of the information that will be included is provided below:
>
> **No. of Distinct Entities per Entity Type**
> Entity Type | Count
> --- | ---
> Symptom | 33,548
> Health Condition | 8,171
> Medicine | 17,862
> Specialist | 7,024
> Age | 5,585
> Dosage | 8,006
> Medical Procedure | 3,752
>
> **Average Lengths (No. of Tokens) per Entity Type**
>
> Entity Type | Average No. of Tokens
> --- | ---
> Symptom | 3.96
> Health Condition | 2.33
> Medicine | 2.43
> Specialist | 2.09
> Age | 1.90
> Dosage | 4.91
> Medical Procedure | 2.37
>
> **Regarding tagging format**: We also briefly considered annotating the offsets, but as the reviewer has pointed out, our own examination of past work also suggested that this is not a commonly chosen approach, which is why we ultimately chose to not pursue it for now. In our humble opinion, the rarity of this approach combined with its apparent importance in languages such as Bengali is indicative of the need for further research in this area.

---

### Official Review · Reviewer_V5Uk · 2023-08-04

**Soundness:** 3

**Excitement:**

4: Strong: This paper deepens the understanding of some phenomenon or lowers the barriers to an existing research direction.

**Missing References:**

Conway M, Hu M, Chapman WW. Recent Advances in Using Natural Language Processing to Address Public Health Research Questions Using Social Media and Consumer Generated Data. Yearb Med Inform. 2019 Aug;28(1):208-217. doi: 10.1055/s-0039-1677918. Epub 2019 Aug 16. PMID: 31419834; PMCID: PMC6697505.

Deléger L, Campillos L, Ligozat AL, Névéol A. Design of an extensive information representation scheme for clinical narratives. J Biomed Semantics. 2017 Sep 11;8(1):37.

Neves M, Ševa J. An extensive review of tools for manual annotation of documents. Brief Bioinform. 2021 Jan 18;22(1):146-163.

Nils Reimers and Iryna Gurevych. 2017. Reporting Score Distributions Makes a Difference: Performance Study of LSTM-networks for Sequence Tagging. In Proceedings of the 2017 Conference on Empirical Methods in Natural Language Processing, pages 338–348, Copenhagen, Denmark. Association for Computational Linguistics.

also G Gonzalez et al. : Social Media Mining for Health (#SMM4H) shared tasks since 2016

**Paper Topic And Main Contributions:**

The manuscript reports on the development of a dataset for the evaluation of named entity recognition for the biomedical field in Bengali. The study corpus is obtained from an online health platform. Seven entity types are covered. Annotations were produced by a team of 20 annotators with substantial agreement. The annotated corpus is then used in an evaluation of masked language models (multilingual as well as Bengali).

**Questions For The Authors:**

Was inter-annotator agreement computed per entity type? Based on the definitions, it can be expected that agreement would be higher for e.g. "age" or "medication" vs. "symptom" or "health condition".

**Reasons To Accept:**

- The manuscripts presents a new resource for medical named entity recognition in Bengali; the annotated corpus as well as annotation guidelines will be shared with the community.
- The dataset is used to evaluate several masked language models (multilingual as well as Bengali)
- An error analysis of the evaluation is reported

## edited to add: ##
The authors' responses to reviewers comments help clarify a number of points and suggest that a revised version as outlined by the response would be improved over the initial submission and likely of interest to the community.

**Reasons To Reject:**

- The annotation process needs to be described in further details. For example:
	* The definitions of the entities in Table 2 seem rather vague. In particular, what is the difference between "symptom" and "health condition"? Was anchoring to a knowedge source such as the UMLS considered? Were other annotated corpus and annotation guidelines considered? A number of annotation schemes for biomedical text are compared and discussed in Deléger et al. 2017; this can be used as a pointer to existing annotated corpora in the biomedical domain and to provide some context to the work presented in terms of entities selected and definitions used.
	* was an annotation tool used? Which one? How was it selected? I recommend refering to Neves and Seva 2019 for an in-depth review of annotation tools; l.201-204 seems to describe the issue of discontinuous entity, which as been addressed in English and other languages. Some annotation tools such as BRAT do support the production of discontinuous entities - as well as nested entities.
	* l. 186 suggests that pre-annotations were supplied to the annotators. How were the pre-annotations produced? Was the pre-annotation quality evaluated, e.g. by comparing to final annotations?
- lines 53-57 seem to describe the general field of NLP for social media and user generated content which has been extensively studied in the past decades for the biomedical field. Example of relevant work can be found in Conway et al. 2019 or Gonzalez et al. Much of this work has addressed English, but some other languages too.
- the differences in results reported for the different models in Table 3 seem very close. Are they significant? Are the results averaged over several runs? Please see Reimer and Gurevitch 2017 on why this might be needed.
- The manuscript does not report information on the data use agreement / user consent that was obtained during the data collection process. Similarly, licence / data use agreement for the corpus that will be shared is missing.
- The process described lines 326 - 332 seems consistent with de-identification (removal of directly identifying information) rather than anonymization (formal guarantee that the link between data and patient cannot be recovered by any means).
- The report of compute use in appendix A.1 could be complemented by CO2 impact estimates, which can be obtained from e.g. http://calculator.green-algorithms.org/  without re-running experiments.

**Reproducibility:**

3: Could reproduce the results with some difficulty. The settings of parameters are underspecified or subjectively determined; the training/evaluation data are not widely available.

**Reviewer Confidence:**

5: Positive that my evaluation is correct. I read the paper very carefully and I am very familiar with related work.

**Typos Grammar Style And Presentation Improvements:**

Conway M, Hu M, Chapman WW. Recent Advances in Using Natural Language Processing to Address Public Health Research Questions Using Social Media and Consumer Generated Data. Yearb Med Inform. 2019 Aug;28(1):208-217. doi: 10.1055/s-0039-1677918. Epub 2019 Aug 16. PMID: 31419834; PMCID: PMC6697505.

Deléger L, Campillos L, Ligozat AL, Névéol A. Design of an extensive information representation scheme for clinical narratives. J Biomed Semantics. 2017 Sep 11;8(1):37.

Neves M, Ševa J. An extensive review of tools for manual annotation of documents. Brief Bioinform. 2021 Jan 18;22(1):146-163.

Nils Reimers and Iryna Gurevych. 2017. Reporting Score Distributions Makes a Difference: Performance Study of LSTM-networks for Sequence Tagging. In Proceedings of the 2017 Conference on Empirical Methods in Natural Language Processing, pages 338–348, Copenhagen, Denmark. Association for Computational Linguistics.

also G Gonzalez et al. : Social Media Mining for Health (#SMM4H) shared tasks since 2016

---

> ### Author Rebuttal · Authors · 2023-08-28
>
> We appreciate that the reviewer took the time to go through our paper thoroughly. We hope to address all of the concerns they have expressed.
>
> **Response to Reasons to Reject #1**: A health condition includes both diseases and medical events. A symptom is an indicator of a health condition. For example, if someone had a bone fracture, the fracture itself would be the health condition while any pain they experience would be the symptom.
>
> Previous work with CHQs under the English language has made use of UMLS specifically [1], and we did examine this work in the initial stages of our research. Unfortunately, the complete lack of medical terminology used in our data along with the informal nature of the language used in the samples meant that we could not determine mappings between the official UMLS terminology and our data to a useful extent.
>
> We did, however, study the terminology quite thoroughly before establishing our entity definitions. We also studied the guidelines of several past medical NER datasets, as well as the ones on CHQ specifically, and used our findings as the basis for our entity definitions.
>
> **Reference**:
>
> [1] Shweta Yadav, Deepak Gupta, and Dina Demner-Fushman. 2022b. Chq-summ: A dataset for consumer healthcare question summarization. arXiv preprint arXiv:2206.06581.
>
> **Annotation Tool**: To meet the different requirements of our data curation process we developed our own customized annotation tool which had a user-friendly UI and support across different platforms. The issue of discontinuous entities does not arise due to the custom-built annotation tool, since there are alternatives that allow for this as the reviewer has pointed out. The decision to not use discontinuous entities was a conscious one. Studying past research, we found that by far the most commonly used format for NER annotations is the IOB format. Since the domain of Bengali medical NER is still in its infancy, we chose to follow this format so as to maximize the usability of our work.
>
> **Pre-annotation policy**: We used pre-annotated samples to calculate the Inter-annotator agreement.  The pre-annotations were produced initially by the authors while establishing the entities to be used. We studied existing datasets [1] and their guidelines extensively when creating these annotations to ensure they are of high quality. For evaluating the final annotation quality, we followed the methodology provided by Kabir et al. (2023).
>
> **Reference**:
>
> [1] Salim Sazzed. 2022. Banglabiomed: A biomedical named-entity annotated corpus for bangla (bengali). In Proceedings of the 21st Workshop on Biomedical Language Processing, pages 323–329
>
> [2] Kabir, Mohsinul, et al. "DEPTWEET: A typology for social media texts to detect depression severities." Computers in Human Behavior 139 (2023): 107503.
>
> **Regarding lines 53-57**: Thank you for this critical observation from the reviewer. We acknowledge that we misspoke here by saying that ‘it is yet to be addressed’.
> We will change the text with relevant references:
>
> Current Statement | Statement for Revised Manuscript
> ---|---
> Being able to extract the correct medical knowledge required to answer the question from the vague descriptions containing unstandardized medical terms used by the general public is a challenging task *that is yet to be addressed* | Being able to extract the correct medical knowledge required to answer the question from the vague descriptions containing unstandardized medical terms used by the general public is a challenging task.
>
>
> **Regarding results reported in Table 3**: The results shown in the paper were not averaged over several runs. We acknowledge that running for multiple trials will provide more statistical evidence on the overall performance of the models, so we have extended our experiments for multiple trials. The following table  shows the results for each of the three language models, mBERT, BanglaBERT and BanglishBERT, averaged over multiple runs. We will add this result in the Appendix section of the revised manuscript.
>
> Model | Accuracy (%) | Precision (%) | Recall (%) | F1-Score (%)
> ---|---|---|---|---
> mBERT | 87.98 +- 0.25 | 49.46 +- 1.15 | 61.90 +- 0.47 | 54.98 +- 0.82
> BanglaBERT | 87.90 +- 0.01 | 48.00 +- 0.38 | 63.66 +- 0.29 | 54.74 +- 0.34
> BanglishBERT | 88.42 +- 0.14 | 50.20 +- 0.62 | 63.65 +- 0.95 | 56.13 +- 0.75
>
> For the BanglishBERT model, which has the best F1-Score (56.13 +- 0.75%), the result remains similar to that in our original manuscript (56.63%). A similar pattern is observed for the other models as well. From this, we can conclude that, although averaging over multiple runs causes the results to vary slightly, it is not to an extent that should cause concern with the original results.
>
> **Regarding the Data-use agreement**: We have made every attempt to ensure that the data collection process followed existing data privacy laws. The scraping of publicly available data from websites is allowed under the data privacy laws of Bangladesh, and the website itself does not forbid any such collection or republishing under their terms.
>
> For sharing the data, we would prefer to allow downloading the data via GitHub under the [CC BY-NC-SA 4.0 license](https://creativecommons.org/licenses/by-nc-sa/4.0/), as this will be the least restrictive for research purposes. However, if the reviewers feel that this could potentially create issues for us in the future, we can require an agreement before access is granted.
>
> **Regarding lines 326-332**: The reviewer has correctly pointed out that our process is more in line with de-identification rather than anonymization. We can rephrase lines 326 - 332 so as to not imply a formal guarantee of anonymization, since we do not provide proof of such.
>
> **Regarding CO2 impact**: We appreciate the suggestion and will include the results of the CO2 impact estimates as an appendix in our paper.
>
> **Regarding the Inter-Annotator Agreement**: We agree that reporting the inter-annotator agreement per entity type would provide further insights, and we also expect the results to be higher for certain classes than for others. We appreciate that this shortcoming was pointed out and will include the missing information in our paper. However, it should be noted that both the F1-score and Cohen's Kappa take class imbalance into account.

---

### Official Review · Reviewer_LNMK · 2023-08-06

**Soundness:** 3

**Excitement:**

3: Ambivalent: It has merits (e.g., it reports state-of-the-art results, the idea is nice), but there are key weaknesses (e.g., it describes incremental work), and it can significantly benefit from another round of revision. However, I won't object to accepting it if my co-reviewers champion it.

**Paper Topic And Main Contributions:**

This paper primarily addresses the problem of insufficient and inadequate datasets for Named Entity Recognition (NER) in the Bengali language, particularly in the context of Consumer Health Questions (CHQs). This limitation hinders the development of effective decision-making systems for medical applications intended to serve Bengali-speaking populations. The problem is complex due to the inherent flexibility and diversity in the Bengali language, along with its widespread informal usage and significant regional dialectal variations.

The main contributions of this paper are:

New Data Resource: The paper introduces 'Bangla-HealthNER,' a manually annotated Bengali medical NER dataset. With 31,783 samples, it is the largest available Bengali medical NER dataset. This dataset is gathered from a public online health platform in Bangladesh, ensuring a broad representation of day-to-day language usage in various linguistic styles and regional dialects. This contribution is particularly significant as it helps fill the gap for low-resource languages.

Real-world Relevance: This dataset, given its source, incorporates the informal language commonly used by patients, adding a layer of realism and practical applicability that could prove vital to the development of medical decision-making systems.

Evaluation and Benchmarking: The dataset has been evaluated on state-of-the-art Bengali and multilingual token classification models to establish a benchmark. BanglishBERT achieved the highest performance with an F1-score of 56.63%, highlighting the inherent difficulty and complexity of the task.

Public Availability: The authors intend to make the 'Bangla-HealthNER' dataset publicly available upon acceptance of the manuscript, fostering future research and development in this area.

Overall, this paper contributes to the field of NLP in a low-resource language context, by providing a dataset and benchmarks for NER in Bengali medical language processing.

**Reasons To Accept:**

The constructive work of providing experimental resources for the Medical NER task in the low-resource language of Bengali is good, but the workload and contributions are not sufficient.

**Reasons To Reject:**

While this research work appears to be primarily resource-oriented, the authors have made commendable contributions by undertaking substantial annotation tasks to enrich the language resources. However, several issues remain:

Insufficient Benchmark Results: The inclusion of only GPT-3.5 and BERT benchmark results is inadequate. The authors should provide more solid comparative experiment details. There is a lack of information in the code section, making reproduction challenging. Please include more details to validate the generalizability of the research results or findings in broader contexts or applications.

Data Set Detail and Experimentation: More information about the dataset and broader experimental results on large language models and downstream task models are required. Also, consider augmenting this language dataset with samples from other language resources. This could help enhance the completeness of both the dataset and the corresponding research report.

**Reproducibility:**

3: Could reproduce the results with some difficulty. The settings of parameters are underspecified or subjectively determined; the training/evaluation data are not widely available.

**Reviewer Confidence:**

4: Quite sure. I tried to check the important points carefully. It's unlikely, though conceivable, that I missed something that should affect my ratings.

---

> ### Author Rebuttal · Authors · 2023-08-28
>
> We are grateful to the reviewers for their kind reviews. We hope to update the final version of our manuscript to address their expressed concerns and are including our responses herein.
>
> ### Response to Reasons to Reject #1:
>
> **Insufficient Benchmark**: The number of models available that are capable of performing token classification on Bengali text is limited. There are only three language models that are widely used in the domain, BanglaBERT, BanglishBERT, and mBERT, all three of which have been included in our benchmark.
>
> **Code Reproduction**: We have provided details regarding the experimental setup and hyper-parameters in Appendix A1. Moreover, all the codes related to dataset curation, preprocessing, post-processing, baseline training, etc. will be publicly available upon the acceptance of the manuscript. We will put extra effort into improving the documentation of the code-base as much as possible to ensure reproduction.
>
> **Generalizability and broader contexts**: The research presented in this paper has implications that extend beyond its specific focus on Named Entity Recognition (NER) in Consumer Health Questions (CHQs). The challenges addressed by the curated dataset, such as dealing with informal language, complex sentence structures, and regional dialects, are observed in other downstream tasks for other languages as well. As a result, the findings of this research can be generalized to broader contexts and applications.
>
> This dataset can be leveraged to achieve Cross-Domain Adaptability in the fields such as social media, customer reviews, and online forums, where users communicate using colloquial and unstructured language. Regarding Cross-Linguistic Transferability, the proposed insights are transferable to other languages with similar linguistic characteristics. This could prove beneficial in developing NER models for languages with limited labeled data. The adaptability of NER as a feature might facilitate automated Clinical Decision-making systems as well.
>
> ### Response to Reasons to Reject #2:
>
> **Venue Suitability**: We thank the reviewer for the valuable suggestion. Also, we would like to bring to the reviewer's kind attention that the resources track of EMNLP is a diverse platform that makes room for a variety of tasks and has accepted similar kinds of resources on under-resourced languages such as Bengali in the past [1,2]. If accepted, our work has the potential to contribute significantly to the Clinical Health community and the exposure of such a venue like EMNLP would help our work reach its stakeholders.
>
> **References**:
>
> [1] Ekram, Syed Mohammed Sartaj, et al. "BanglaRQA: A Benchmark Dataset for Under-resourced Bangla Language Reading Comprehension-based Question Answering with Diverse Question-Answer Types." Findings of the Association for Computational Linguistics: EMNLP 2022. 2022. (https://aclanthology.org/2022.findings-emnlp.186/)
>
> [2] Islam, Khondoker Ittehadul, et al. "SentNoB: A dataset for analyzing sentiment on noisy Bangla texts." Findings of the Association for Computational Linguistics: EMNLP 2021. 2021. (https://aclanthology.org/2021.findings-emnlp.278/)
>
> ### Response to Reasons to Reject #3:
>
> We thank the reviewer for this valuable insight. For augmenting a language dataset, we can follow approaches such as context-based or rule-based augmentation. Since this is an NER task, context-based augmentation requires re-annotating the dataset which is not feasible in this time frame. However, rule-based augmentation that involves substituting tokens with synonyms could indeed improve the quality of the dataset and is something we will be looking into.

---

### Meta-Review · Area_Chair_mxL2 · 2023-09-14

**Recommendation:** 4

**Metareview:**

The main contribution of this paper consists in the construction of a new manually annotated dataset and a benchmark for NER in Bengali for medical language processing. The experimental evaluation, though limited, appears to be sufficient for assessing quality and justifying the effort.  Reviewers and authors engaged in a constructive discussion, and the authors' responses clarified numerous points and show willingness to incorporate reviewers' suggestions into the next version of the paper. Reviewers finally converged on Soundness and two of found the paper exciting. Overall, this short paper can indeed be of great interest to the EMNLP community.

**Pros**

- coverage of a non-English, less-resourced language (Bengali);

- coverage of a significant domain, biomedicine;

- coverage of ordinary language usage, various linguistic styles, and regional dialects.

- creation of the largest dataset available for Bengali, with and evaluation against SOTA for Bengali;

- findings appear to be generalizable to other dialogical, informal contexts;

- the paper is easy to follow and understand.



**Cons**

- as this paper focuses on a data resource, it lacks essential details regarding the annotation process and dataset composition properties, as pointed out by Reviewers V5Uk and NwnB. Nevertheless, the authors have shown awareness of these missing details, which will be included in the revised version;

- The significance of the performance differences among the various models is not self-evident and would need a more thorough investigation. Authors however performed an extra test and provided the results in the discussion, which clarifies the point and should be included in the paper or appendix;

- some background references need to be acknowledged (see reviewers' suggestions);

---

### Decision · Program_Chairs · 2023-10-07

**Decision:**

Accept-Findings

**Comment:**

The main contribution of this paper consists in the construction of a new manually annotated dataset and a benchmark for NER in Bengali for medical language processing. The experimental evaluation, though limited, appears to be sufficient for assessing quality and justifying the effort.  Reviewers and authors engaged in a constructive discussion, and the authors' responses clarified numerous points and show willingness to incorporate reviewers' suggestions into the next version of the paper. Reviewers finally converged on Soundness and two of found the paper exciting. Overall, this short paper can indeed be of great interest to the EMNLP community.

**Pros**

- coverage of a non-English, less-resourced language (Bengali);

- coverage of a significant domain, biomedicine;

- coverage of ordinary language usage, various linguistic styles, and regional dialects.

- creation of the largest dataset available for Bengali, with and evaluation against SOTA for Bengali;

- findings appear to be generalizable to other dialogical, informal contexts;

- the paper is easy to follow and understand.



**Cons**

- as this paper focuses on a data resource, it lacks essential details regarding the annotation process and dataset composition properties, as pointed out by Reviewers V5Uk and NwnB. Nevertheless, the authors have shown awareness of these missing details, which will be included in the revised version;

- The significance of the performance differences among the various models is not self-evident and would need a more thorough investigation. Authors however performed an extra test and provided the results in the discussion, which clarifies the point and should be included in the paper or appendix;

- some background references need to be acknowledged (see reviewers' suggestions);